# Influence of Martensite/Bainite Dual Phase-Content on the Mechanical Properties of EA4T High-Speed Axle Steel

**DOI:** 10.3390/ma16134657

**Published:** 2023-06-28

**Authors:** Yan Zhang, Yu Cao, Guangjie Huang, Yanyang Wang, Qilei Li, Jie He

**Affiliations:** 1College of Materials Science and Engineering, Chongqing University, Chongqing 400044, China; mgzhangyan@163.com (Y.Z.); yucao928@cqu.edu.cn (Y.C.); liqilei1122@foxmail.com (Q.L.); m19132053328@163.com (J.H.); 2Baowu Group Masteel Rail Transit Materials Technology Company Limited, Ma’anshan 243000, China; 3Maanshan Iron and Steel Co., Ltd., Ma’anshan 243000, China; 18395577957@163.com

**Keywords:** EA4T, lath martensite, granular bainite, mechanical properties, prediction model

## Abstract

In this work, we have investigated the effect of martensite/bainite dual phase content on the mechanical properties of EA4T high-speed axle steel. For evaluation and control of the strength, ductility, and toughness of steel, the microstructure of lath martensite (LM) and granular bainite (GB) was clarified through an optical microscope (OM), electron backscatter diffraction (EBSD), and transmission electron microscopy (TEM). Besides, the tensile fracture morphology was studied by scanning electron microscopy (SEM). For this purpose, this study conducted a quantitative analysis of the LM and GB fractions using the Pro Imaging software-2018 of OM. The remarkable effect of the LM/GB structure on mechanical properties is discussed. The results have shown that by increasing the volume fraction of the GB structure, the LM structure is refined and its microhardness and strength are improved. Meanwhile, the micro strength of LM follows the Hall–Petch relationship with the lath martensite packet size. Subsequently, the mechanical property prediction model of EA4T steel based on the LM/GB content was established by regression analysis of all experiment dates. When the LM fraction in the steel is about 40–70%, a superior combination of strength, ductility, and toughness can be obtained in EA4T steel.

## 1. Introduction

In the past few decades, EA4T steel has been widely used to produce high-speed axles due to its excellent combined properties of strength, ductility, and toughness [1,2,3]. With the rapid development of high-speed railway systems, the ambient temperature of trains in service can be lower than −50 °C in some cold regions. Thus, EA4T axle steel is subjected to a double challenge of different axle loads and extremely cold conditions in service, and the axle fracture caused by structure defects can lead to some potentially disastrous consequences [4,5]. Zhu et al. studied the microstructure and properties of EA4T axle steel in EA200 and EA380 high-speed trains, and their results clearly showed that the microstructure is mainly composed of tempered lath martensite (LM) and granular bainite (GB) structures, with the yield strength lower than 550 MPa and the ductile-brittle transition temperature (DBTT) as low as −38 °C [6]. Therefore, it is important to reveal the influence mechanism of LM/GB structures on the mechanical properties of EA4T steel, which has a significant effect on improving the comprehensive properties of axle steels.

EA4T steel has a tempered martensite and bainite structure, allowing trace ferrite to be treated by quenching-tempering [1]. For industrial steels, the traditional view is that the martensite structure has high strength, ductility, and toughness [7,8,9,10]. Meanwhile, the typical bainite structure also plays an important role in its mechanical properties [11]. Tomita et al. have pointed out that the martensite and bainite dual phase structure can significantly improve the strength and toughness of steel [11,12,13,14,15,16,17,18,19,20,21,22,23,24,25,26,27,28]. In previous works, martensite is usually described in terms of the crystallographic features of packet, block, or lath-like structures [9,29,30]. Besides, granular bainite is an island structure with austenite or other transformation phases distributed within the ferrite matrix [21,22,23,24]. Granular bainite is mainly composed of lath morphology [23]. George et al. indicated that the yield strength is in Hall–Petch relation to the martensite packet size in 25CrMo48V dual phase steels [29,30]. When the bainite volume fraction increases, the martensite packet size becomes smaller, thus leading to a higher martensitic micro strength [31]. On the other hand, Katsumate et al. have found that the impact toughness of martensite/bainite dual phase structure in low-carbon steels is better than that of single martensitic steels, while its influence on strength is very little [27,28,32,33]. Under the impact load, the bainite structure produces a refinement of the martensite matrix, which can also increase the resistance against brittle fracture [27]. Besides, the dual phase microstructure is prone to enhance the crack initiation energy and crack propagation energy, which consequently improves the low-temperature toughness [33,34]. Moreover, some previous researchers have also paid attention to dealing with the relationship between microstructure evolution and mechanical properties. Samuel et al. discovered that a combination of superior strength and toughness can be obtained in medium carbon low alloy steels with 45% and 65% martensite volume fractions compared with that of single martensite or granular bainite steel [23]. Tomita et al. indicated that a synergetic enhancement of strength, ductility, and impact toughness can be achieved in the 0.40C-Ni-Cr-Mo steel after quenching with subsequent tempering when the ratio of martensite to bainite is about 3:1 [24,25]. For the previous investigation on martensite-bainite steel, Chen et al. found that some excellent mechanical properties can be gained when the volume fraction of bainite is controlled to be 50–60% [35,36]. Simultaneously, Smoljan et al. established the prediction model of mechanical properties for 41Cr4 and 42CrMo4 steel by analyzing the proportion of martensite and bainite phases [37,38]. Although EA4T grade steel is widely used as a typical Cr-Mo alloy structure steel, few researchers have systematically studied the impact of LM/GB structure on strength and toughness, and there are no reports on the prediction model for mechanical properties of EA4T steel based on microstructure evolution as well.

In this work, the relationship between microstructure and mechanical properties in EA4T steel was entirely investigated. Firstly, the samples with different volume fractions of LM/GB were quantified and characterized through the OM and EBSD techniques. Meanwhile, microhardness, tensile properties, and impact toughness were also systematically measured with respect to different microstructure characteristics. Afterward, through evaluating the LM/GB volume fraction the prediction models of mechanical properties were established by the regression analysis of experimental data. Finally, the mechanism by which the LM/GB structure influences the strength, ductility, and toughness of EA4T steel was also revealed.

## 2. Experimental Material and Procedure

### 2.1. Material Preparation

The EA4T axle steel was received as the forged bars after the electric furnace melted them in this work. The chemical compositions (in wt%) are listed as follows: C 0.27, Mn 0.71, Cr 1.08, Si 0.29, Mo 0.24, Ni 0.25, V 0.037, Cu 0.03, P 0.009, S 0.002. The detailed preparation process is introduced as follows: ∅600 mm continuous casting billet→280 mm × 280 mm billet→∅240 mm forging bar, and subsequent heat treatment. The heat treatment process mainly consists of the following sequential steps: (i) quench heating at 880 °C for 5 h, (ii) quenching in water for 20 min, and (iii) tempering at 640 °C for 7 h (see Figure 1).

The as-received microstructure of EA4T steel is primarily composed of martensite and bainite. The LM/GB volume fractions at different positions in the cross-section are diverse because the cooling rate gradually decreases from the surface to the inside part of the forged bar during the quenching process, and water was used as the quenching medium. As shown in Figure 2a, the austenite grain size is greater than 8.0, and the volume fraction of ferrite is less than 3% from the observation of optical microscopy (see Figure 2b). The ferritic content in the most severe field of view of the whole test surface of the sample is observed with the help of the microscope eyepiece, usually with a magnification of 500×.

### 2.2. Investigation of Microstructure and Hardness

The microstructure was observed by a Carl Zeiss Axio Imager A2m optical microscope (Zeiss, Oberkochen, Germany), and 12 metallographic specimens were prepared at different positions in the cross-section of the experiment steel. The proportion of LM and GB structures in EA4T steel was quantified by Pro Imaging software-2018, and the average value was the final quantified result of the martensitic/bainitic volume fraction. Min-Seok Baek et al. described the OP method to quantify the fraction of martensite-bainite steel [36,39]. In addition, the color metallographic infection method was used to verify the accuracy of the quantitative results. The proportion of color metallographic reagent is as follows: sodium thiosulfate, 24 g; cadmium chloride, 2–2.5 g; citric acid, 3 g; and water, 100 mL. Additionally, the crystallography characteristics were further evaluated on a JEOL JSM-7800F field emission scanning electron microscope (SEM) (JEOL, Tokyo, Japan) equipped with an electron backscatter diffraction (EBSD) detector (Oxford Instruments, Abingdon, UK). Finally, the corresponding EBSD mapping data was analyzed using OIM Analysis software (TSL OIM 7.3).

Macrohardness values with different LM/GB volume fractions were measured according to the ASTM E92-17 at a load of 10 kg using the Wison VH3100 semi-automatic Vickers hardness machine (Wison, Houston, TX, USA). Meanwhile, microhardness measurements were also done on the martensite and bainite phases using a load of 1 kg. In the present work, the macrohardness value was obtained by calculating an average of five testing points, while the mean microhardness value was based on three testing points. In this paper, the hardness model proposed by Maynier [40] was used to calculate the microhardness of martensitic and bainitic structures after quenching and tempering. The chemical composition was also obtained by a spectral method, and the results of the three samples are shown in Table 1.

### 2.3. Tensile and Impacting Tests

Cylindrical tensile specimens with a 50 mm gauge length and 10 mm diameter were fabricated from the experiment steel. The tensile test was carried out using a Zwic 600 w material machine at a constant cross-head speed of 6.7 × 10^−3^ s^−1^ according to the ISO 6892 standard-2009. Charpy U-nouth (CUN) and Charpy V-nouth (CVN) impact specimens with a size of 10 mm × 10 mm × 55 mm were machined. The CUN impact properties were tested on a pendulum impact ZBC752N-3 machine. In addition, the transition temperature of impact toughness with well-matched strength and ductility was measured at test temperatures ranging from −80 to 20 °C, and the specimens were 5 mm CUN and 2 mm CVN, respectively. Two low-temperature impact test methods can clarify the test results more accurately. To estimate the ductility of experimental steel with different martensite and bainite volume fractions, a series of SEM fractography images were obtained from the fractured surfaces of tensile specimens.

## 3. Results

### 3.1. Microstructure and Hardness of LM-GB Steel

As shown in Figure 3a, the inverse pole figure (IPF) of LM-GB steel indicates that the orientation characteristics of adjacent block structures are different. Besides, we can also observe some lath structures with slight misorientation inside the martensite block. Generally, the image quality (IQ) value denotes the sharpness of Kikuchi patterns at a given scanning point. Thus, the IQ map can represent not only the real microstructure but also the integrity of the crystal structure; namely, a lower IQ value always corresponds to a severely distorted matrix with plentiful deformation substructures. The contrast of the gray IQ map in Figure 3b can reflect the distribution of different phases since the lath martensite is mainly composed of large dislocations, which are inline to display a dark contrast in this IQ map. The distribution of grain boundaries with different rotation angles (i.e., low-angle grain boundary, medium-angle grain boundary, and high-angle grain boundary) can be observed in Figure 3c, which expressly signifies that the block structures are mainly surrounded by high-angle grain boundaries, with their number fraction being only 34.2%.

Figure 4 shows the typical optical micrographs with martensite fractions of 24.9%, 50.5%, and 74.9% in LM-GB steel. The white and black areas (see Figure 4a) are the tempered martensite and bainite structures, respectively. According to Figure 4a, a color metallographic method was also used to verify the volume fraction of different phases, the dark area (blue) is the martensite, and the light area (white and yellow) is the bainite (Figure 4b). The quantified results of Figure 4b were consistent with those of Figure 4a. In this work, the volume fraction of martensite and bainite was analyzed by metallurgical quantitative software-2018, and twelve combinations of LM/GB structure can be obtained with martensitic fractions of 24.1–74.9% and bainitic fractions of 25.1–75.9%.

Microhardness measurements showed that the martensitic hardness values are 320, 274, and 237 HV1, while the bainitic hardness values are 204, 217, and 214 HV1 for the above three experiment steels, as shown in Figure 4. With the increase in bainite fraction, the characteristics of martensite and bainite become clearer. The LM structure is lath-like, and the lath bundle and its orientation can be observed (denoted by the red dotted line in Figure 4a,b), and the GB structure is lath-like or plate-like in shape (denoted by the green dotted line in Figure 4a,b). On the other hand, granular bainite consists of massive areas of ferrite in the neighborhood of granules of bainite laths, or small white particles.

### 3.2. Effect of Martensitic/Bainitic Dual Phase Structure on the Hardness

The effect of martensite and bainite structures on the mechanical properties of EA4T steel has been systematically studied. Considering the mechanical properties of hardened steel, the hardenability of steel is mainly related to its chemical composition. Maynier et al. have developed some regression equations to determine the critical cooling rate of martensite and bainite based on the composition features (Equations (1)–(3)) [40] and also predicted the hardness of different phases based on the cooling rate and chemical composition (Equations (4)–(6)) [40].

The quenching microstructure has a genetic effect on the final properties of steel [41]. Therefore, the volume fraction of martensite and bainite after quenching also adopts the quantitative results of the martensite/bainite fraction in this work. *HV_E_*, *HV_E-M_*, and *HV_E-B_* can represent the macrohardness of steel and the microhardness of martensite and bainite after quenching, respectively. While *HV_T_*, *HV_T-M_*, and *HV_T-B_* correspond to the macrohardness of steel and the microhardness of martensite and bainite after quenching and tempering, respectively. Besides, *V_M_* and *V_B_* are the critical cooling rates of martensite and bainite in °C/h, respectively. *P_a_* is the austenitizing parameter, which can be determined by involving time (*t*) and temperature (*T*). As shown in Table 1, the quenched microhardness of martensite and bainite is 532 ± 15 *HV*1 and 326 ± 5 *HV*1, and the quenched hardness range of specimens is 371–547 *HV* according to the microhardness and volume fraction of the martensite/bainite. Besides, the microhardness range of tempered martensite and bainite is 230–330 *HV*1 and 196–223 *HV*1, and the tempered hardness range of specimens is 219–247 *HV*10.
(1)$$logV_{M}=9.81-(4.62C+1.1Mn+0.54Ni+0.50Cr+0.66Mo+0.00183P_{a})$$
(2)$$logV_{B}=10.17-(3.80C+1.07Mn+0.70Ni+0.57Cr+1.58Mo+0.0032P)$$
(3)$$P_{a}=t\times T$$
(4)$$HV_{M}=127+949C+27Si+11Mn+8Ni+16Cr+21(logV_{M})$$
(5)$$HV_{B}=-323+185C+330Si+153Mn+65Ni+144Cr+191Mo+\;logV_{B}\times (89+53C-55Si-22Mn-10Ni-20Cr-33Mo)$$
(6)$${HV}_{E}=\sum_{i}{fi\left(HV\right)}_{i}$$

The variation in hardness with different microstructure contents is shown in Figure 5. The quenched microhardness of martensite/bainite is closely related to its chemical composition and critical cooling rate. The microhardness of tempered martensite decreases with the increase of martensite fraction (see Figure 5a), while the microhardness of bainite is less affected by the martensite fraction (see Figure 5b). On the other hand, the martensite microhardness and the macrohardness of specimens after quenching andtempering are also affected by the bainite fraction (see Figure 5d). As the bainite content increases, the microhardness of martensite increases, and macrohardness of steel decreases, which is an important discovery in this investigation. When the volume fraction of bainite is 75.9%, 49.5%, and 25.1%, the microhardness of the tempered martensite is 320 *HV*1, 274 *HV*1, and 237 *HV*1, respectively. When the bainite fraction is 24.1%, the martensitic hardness is close to bainitic hardness (see Figure 5d).

The microhardness of martensite is always higher than that of bainite; that is, the hardness of steel is more affected by the volume fraction of martensite. With the rising martensite fraction, the hardness of steel increases, which mostly accords with the linear regression in Figure 5c. Hence, Equation (6) was used to calculate the hardness of LM-GB steel after quenching and tempering (*HV_T-C_*), which is also evenly distributed on the regression curve of tempering hardness (*HV_T_*). The average error between the *HV_T-C_* and *HV_T_* is 4.82%, and the tempering hardness also conforms to the Maynier hardness model.

### 3.3. Effect of Martensite/Bainite Dual Phase Structure on Mechanical Properties

To determine the optimum volume fraction of LM and GB for improving the comprehensive mechanical properties of LM-GB steel, all the testing results with various volume fractions of martensite and bainite are presented in Table 2.

The quenching hardness and the tempering hardness of specimens have a linear relationship with the martensite fraction (see Figure 5c), and the strength properties and the tempering hardness also show a linear relationship, as depicted in Figure 6a. Simultaneously, the relationship between hardness and strength can also be obtained based on the results in Table 2, which can be expressed as *R_p_*_0.2_ = (2.5–2.7) × *HV_T_*, *R_m_* = (3.1–3.3) × *HV_T_*, respectively. Therefore, the yield and tensile strength are strongly related to the volume fraction of martensite, which is distinctly shown in Figure 6b. Besides, it can be observed from Figure 6c,d that the variation of elongation (EL), reduction of area (Z), and impact energy with the martensite fraction all show an increasing tendency with a rising martensite fraction with a percentage less than 50%. The above results of tensile strength are consistent with the previous work from Abbaszadeh et al. [11], and those of impact toughness are similar to the results from investigations conducted by Tomita et al. [24].

According to the results in Table 2, the combination of strength, ductility, and impact toughness is superior for the specimens with a martensite volume fraction of 44.5% to 50.5%. The low-temperature impact toughness was also studied, and the ductile-brittle transition temperature (DBTT) curves of CUN and CUV were fitted by the Boltzmann function as presented in Figure 7, which indicates the transition temperature of toughness to brittleness is −57–62 °C. Tomita et al. [24,25] believed that the martensite/bainite structure has a higher toughness, which is mainly affected by the volume fraction of each phase in the steel. Microcraks are generally initiated at one bainite area and finally stopped by the author’s bainite area. Due to the influence of the martensite/bainite interface, initiation cracks are likely to change direction at the interface and absorb more energy, thus reducing the 50% DBTT.

### 3.4. Prediction Model of Mechanical Properties

Mechanical properties with different volume fractions of martensite and bainite were analyzed mathematically, and a prediction model of mechanical properties related to the microstructure evolution was established.

The regression results between the quenching/tempering hardness and the martensite/bainite volume fractions are exhibited in Equations (7) and (8). The mechanical properties are largely determined by the hardenability [42], which can be characterized by the microstructure evolution. The tensile strength, elongation, and impact toughness are all related to the hardness [38]. Therefore, we likewise performed a series of regression analyses between mechanical properties and hardness, as shown in Equations (9)–(13).
(7)$${HV}_{E}=462.84+0.00478\times {\left(M\%\right)}^{2}-0.0167\times {\left(B\%\right)}^{2}$$
(8)$${HV}_{T}=222.67+0.00431\times \left(M\%\right)+0.00039\times \left(B\%\right)$$
(9)$$R_{m}=713.2259+0.9394\times {HV}_{E}-0.0007\times {\left({HV}_{E}\right)}^{2}-3.7747\times {HV}_{T}+0.009\times {\left({HV}_{T}\right)}^{2}$$
(10)$$R_{p0.2}=1013.557+2.7766\times {HV}_{E}-0.0031\times {\left({HV}_{E}\right)}^{2}-8.8799\times {HV}_{T}+0.0218\times {\left({HV}_{T}\right)}^{2}$$
(11)$$A=-98.0367+0.3911\times {HV}_{E}-0.0005\times {\left({HV}_{E}\right)}^{2}+0.2979\times {HV}_{T}-0.0006\times {\left({HV}_{T}\right)}^{2}$$
(12)$$Z=793.222+2.0204\times {HV}_{E}-0.0023\times {\left({HV}_{E}\right)}^{2}-9.9072\times {HV}_{T}+0.02116\times {\left({HV}_{T}\right)}^{2}$$
(13)$${AKU}_{2}=-1697.75+2.2302\times {HV}_{E}-0.0026\times {\left({HV}_{E}\right)}^{2}+11.1026\times {HV}_{T}-0.0237\times {\left({HV}_{T}\right)}^{2}$$

Meanwhile, the relationship between mechanical properties and martensite/bainite volume fractions was also established, as illustrated in Equations (14)–(18). According to the above equations, it can be inferred that the influence slopes of the martensite and bainite fractions on the tensile strength are 0.00124 and 0.00772, respectively. While the influence slopes of martensite and bainite fractions on the yield strength are 0.00246 and 0.00703, respectively. According to Equation (18), the influence slopes of the martensite and bainite fractions on the impact toughness are 0.00696 and 0.0078, respectively. When the volume fractions of martensite and bainite are closer to each other, the impact toughness is excellent.
(14)$$R_{m}=770.9941+0.00124\times {\left(M\%\right)}^{2}-0.00772\times {\left(B\%\right)}^{2}$$
(15)$$R_{p0.2}=613.8447+0.00246\times {\left(M\%\right)}^{2}-0.00703\times {\left(B\%\right)}^{2}$$
(16)$$A=28.28597-0.00098\times {\left(M\%\right)}^{2}-0.0012\times {\left(B\%\right)}^{2}$$
(17)$$Z=94.6056-0.0039\times {\left(M\%\right)}^{2}-0.00497\times {\left(B\%\right)}^{2}$$
(18)$${AKU}_{2}=126.2119-0.00696\times {\left(M\%\right)}^{2}-0.0078\times {\left(B\%\right)}^{2}$$

To verify the accuracy of the prediction model for the mechanical properties of EA4T steel, the correlation coefficient (*ρ*) and the average absolute relative error (*δ*) were calculated according to Equations (19) and (20), respectively. Where *X_i_* is the experimental data, *Y_i_* is the predicted data, and *N* is the number of data points. X and Y represent the mean values of *X_i_* and *Y_i_*. After calculation and comparison, the *ρ* and *δ* values of the above models were shown in Table 3. Therefore, it can be concluded that the models can accurately predict the mechanical properties of EA4T steel.
(19)$$\rho =\frac{\sum_{i=1}^{N}\left(X_{i}-\bar{X}\right)\left(Y_{i}-\bar{Y}\right)}{\sqrt{\sum_{i=1}^{N}{\left(X_{i}-\bar{X}\right)}^{2}\sum_{i=1}^{N}{\left(Y_{i}-\bar{Y}\right)}^{2}}}$$
(20)$$\delta =\frac{1}{n}\sum_{i=1}^{N}\left|\frac{X_{i}-\bar{X}}{X_{i}}\right|$$

## 4. Discussion

### 4.1. Effect of Microstructure Evolution on Tensile Properties

The room temperature tensile test results with various martensite fractions were plotted on a stress-strain curve (see Figure 8a), and the strength and elongation values were listed in Table 2 as well. It can be observed that the tensile strength increases gradually with a rising martensite content, which can be attributed to the fact that the microhardness of martensite is always higher than that of bainite (Figure 5d). For elongation, necking behavior occurred in all specimens, uniform strain and total strain were very clear, and the uniform strain had slight increases with the increase of martensite fraction. As a result, the total strain of a 50.5% martensitic specimen was the best.

The macroscopic fracture morphology was investigated (Figure 8b–d). It was confirmed that the necking behavior occurred at significant levels (50% of total strain) in both specimens. The proportion of the fiber zone is the largest, as is the proportion of radiation in the 50.5% martensitic specimen, and the ductility of steel is excellent. The radiation zone with 74.5% martensite is slightly lower than that with 24.1% martensite, and the proportion of the fiber zone is similar. The ductility of the 74.5% martensitic steel was slightly superior.

In this study, high-magnification observations of the fiber fracture surfaces identified fracture cracks, dimples, and a type of ductile fracture mode. The dimples presented had two forms, so they were described as micro-dimples (denoted by the blue dotted line in Figure 9) and deeper dampers (denoted by the red dotted line in Figure 8). For EA4T steel after quenching and tempering, cracks appear in the fiber fracture zone, which is caused by the grain boundary quenching stress of steel after undergoing large plastic deformation, as shown in Figure 9(a1,b1). The 50.5% martensitic specimen has more intense plastic deformation under tensile stress, and its plasticity is better, as can be observed in Figure 9(a2,b2). The size of the dimples was slightly larger in the 50.5% martensitic specimen compared to the 74.5% martensite specimen in Figure 9(a3,b3). It has previously been shown that when there is greater quantities of bainite in steel, its boundary becomes the focus of stress and strain, leading to faster crack propagation [20,21]. Furthermore, the deeper dimples in the 24.1% martensite steel were coarse and large enough to be observed on the macro tensile fracture graph (marked by the red dotted line in Figure 8b), yet it also shows a lightly lower uniform strain owing to its large number of larger, deeper dimples. This can be interpreted as the uniform strain decreasing with the increase in bainite volume fraction (see Figure 8a). The total strain is affected by the interaction between martensite and bainite. When the volume fractions of martensite and bainite are closer, the total elongation of steel is higher, as shown in Figure 8a.

The strength and ductility of the 50.5% martensite specimen are superior, which can be explained by several features of the fracture surfaces, which are represented in Figure 7 and Figure 8: (i) by raising the volume fraction of the ductile bainite structure appropriately, the soft bainite structure can promote stress relaxation during a tensile test, and further loading is required for propagation of the crack, and finally the specimen fracture occurs; (ii) The refinement of the martensitic substructure contributes to a slight reduction of the tensile strength, but the dual phase interface is more complex; (iii) We have found large numbers of microcracks, and the micromechanism of ductile fracture involves the initiation, growth, and coalescence of microvoids at the fracture surface; the growth of cracks in a shear fracture mode is restricted.

### 4.2. Influence of the Mechanism of Martensilte/Bainite Structure on Mechanical Properties

EA4T steel can have an excellent combination of strength, ductility, and impact toughness. Based on the regression Equations (14)–(18), when the volume fraction of martensite in the specimens is 40–70%, the yield strength, ductility, and impact toughness are 590–620 MPa, ≥22%, and ≥85 J, respectively. It can be seen from Figure 10 that the ductility and impact toughness of the steel is similar and consistent with the volume fractions of the LM and GB structures. This is mainly because increasing the ductility of the matrix microstructure is an effective way to improve the impact toughness of steel, and the matrix with high ductility can absorb more energy and produce deeper dimples on the fracture surfaces [14].

Ankem et al. found that the strength of dual phase steel is not only related to the strength and volume fraction of each phase but also has a connection with the interaction between different microstructures [43,44]. When bainite content increases, martensite packet size becomes refined, and its strength can be improved accordingly. On the contrary, bainite, as a relatively soft phase, has a very low strength effect strength when the volume fraction of the LM and GB structures is within a reasonable range, according to Figure 10, and then plays a role in balancing strength and toughness. When the bainite structure is very small, its strength nearly matches that of the martensite structure [45].

The yield strength of dual phase steel relies on the volume fraction and the yield strength of each phase [41,46]. As mentioned above, the yield strength can be calculated through the concept of the mixtures rule (Equation (21)). The yield strength of the martensite/bainite dual phase structure ($\sigma_{y}^{Mix}$) increases linearly with increasing the volume fraction of martensite, and the main reason is that the martensite strength is higher than the bainite strength [31,45], and this is consistent with the results of this study (in Figure 5b). According to Equation (21), the yield strengths of martensite and bainite structures ($\sigma_{y}^{M}$ and $\sigma_{y}^{B}$) are approximately calculated for three group dates of martensite fraction in Table 2, which were 24.1–36.2%, 47.2–50.5%, and 50.5–74.9%, respectively. The ratio of $\sigma_{y}^{M}$ and $\sigma_{y}^{B}$ was calculated, and that of tempered microhardness was also calculated. With the increasing volume fraction of bainite, the strength and hardness of the martensite microstructure are gradually improved. All the above results are shown in Table 4.
(21)$$\sigma_{y}^{Mix}=\sigma_{y}^{M}\times \left(M\%\right){+\sigma }_{y}^{B}\times \left(1-M\%\right)$$
(22)$$\sigma_{y}^{M}=\sigma_{i}+k\times S_{m}^{-0.5}$$

The martensite packet size decreases as the volume fraction of bainite is increased [31]. According to the Hall–Petch relationship (Equation (22)), *σ_i_* is frictional stress, *k* is constant, and *S_m_* is martensite packet size. The strength of the martensite structure increases by decreasing the martensite packet size. In the process of high-temperature austenite cooling of the experimental steel, the precipitated bainite, which was lath-like or plate-like in shape, divided the original austenite grains into many smaller regions, and then the martensite that transformed was restricted in the smaller zones and therefore had a refined packet size [31]. Therefore, granular bainite plays an important role in the grain refinement of final microstructures in dual phase steel [42]. This can give a reasonable explanation for why the increase in bainite content leads to the refinement and strengthening of the martensite structure. Finally, the reason for the different combinations of strength and toughness can be attributed to the improved mechanical mechanisms based on the interaction and different volume fractions of martensite and bainite.

## 5. Conclusions

In this paper, the EA4T axle steel was used to study the influence mechanism of the martensite/bainite dual phase content on its mechanical properties. Based on the above analysis, the following conclusions can be drawn:(1)The mechanical property prediction model of EA4T steel was established according to the volume fraction of martensite and bainite. The mathematical model can be used to rapidly predict the mechanical properties of EA4T steel. When the volume fraction of martensite in steel is 40–70%, the yield strength is 590–620 MPa, and the impact toughness is ≥85 J, respectively. The combination of strength, ductility, and impact toughness is excellent when the volume fractions of martensite and bainite are closer, the impact toughness is excellent, and the 50% fraction appearance transition temperature is −60 °C.(2)The beneficial influence has been attributed to the improved mechanisms based on the interaction and different volume fractions of martensite and bainite, which could be explained by the following factors: (i) in terms of the rule of mixtures; (ii) the strength of the martensite/bainite dual phase structure increases linearly with increasing the volume fraction of martensite; (iii) the strength of martensite is higher than that of bainite. Simultaneously, bainite, as a relatively soft phase, plays an important role in balancing strength and toughness.(3)The fracture morphology of tensile specimens has shown that the necking behavior of the specimen occurred at significant levels and that the fracture surfaces exhibited a dimple rupture mode. The 50.5% martensitic specimen has more intense plastic deformation under tensile stress, and its dampers are larger and deeper, thus its plasticity is excellent.(4)According to the concept of mixtures rule, the yield strengths of martensite and bainite structures were approximately calculated for three group dates of martensite fraction, which were 24.1–36.2%, 47.2–50.5%, and 50.5–74.9%, respectively. With the increasing volume fraction of bainite, the strength and hardness of the martensite microstructure are gradually improved.

## Figures and Tables

**Figure 1 materials-16-04657-f001:**
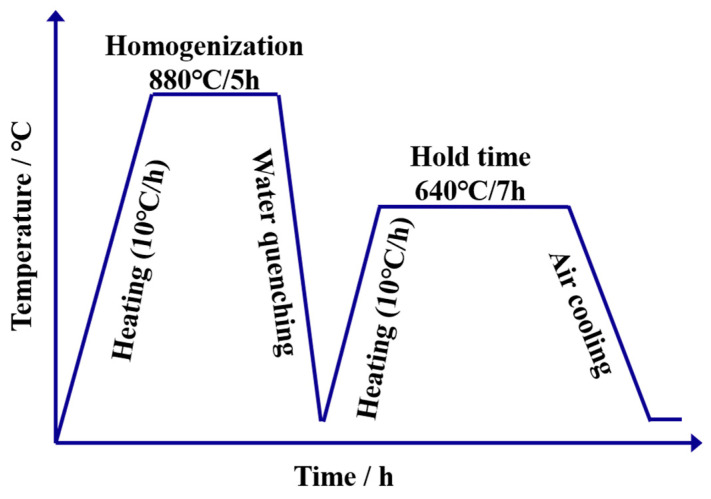
Heat treatment curve.

**Figure 2 materials-16-04657-f002:**
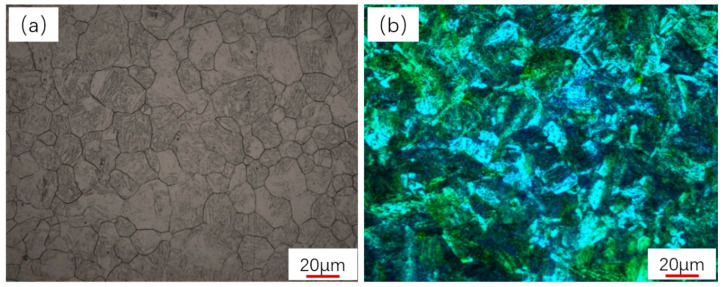
(**a**) Grain size, (**b**) microstructure, LM (blue, dark color), GB (yellow and white et al. light color). OM, typical representative microstructure of the as-revived steel.

**Figure 3 materials-16-04657-f003:**
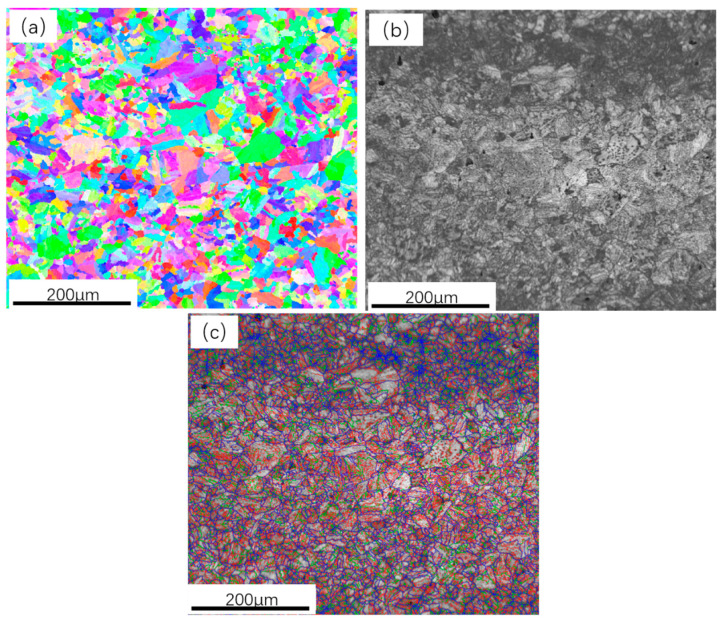
EBSD analysis results of LM-GB steel: (**a**) IPF image, (**b**) IQ image, and (**c**) grain boundary distribution maps.

**Figure 4 materials-16-04657-f004:**
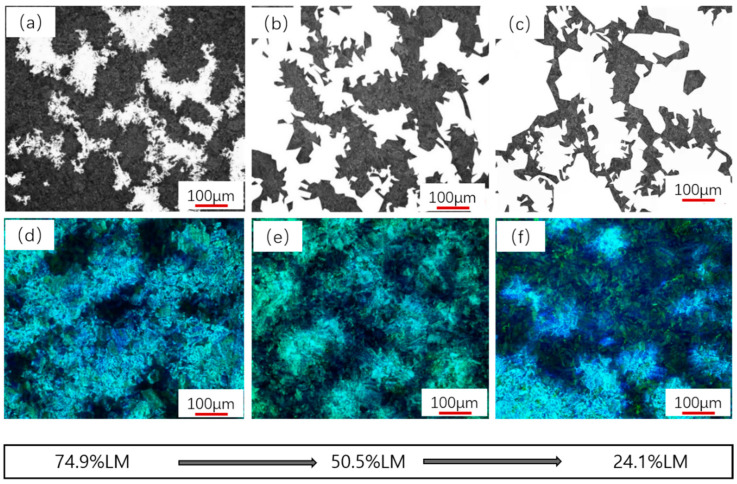
OM images of three LM-GB structures with 24.1%, 50.5%, and 74.9% of the LM structure: (**a**–**c**) metallography quantitative result, (**d**–**f**) metallography tint etching.

**Figure 5 materials-16-04657-f005:**
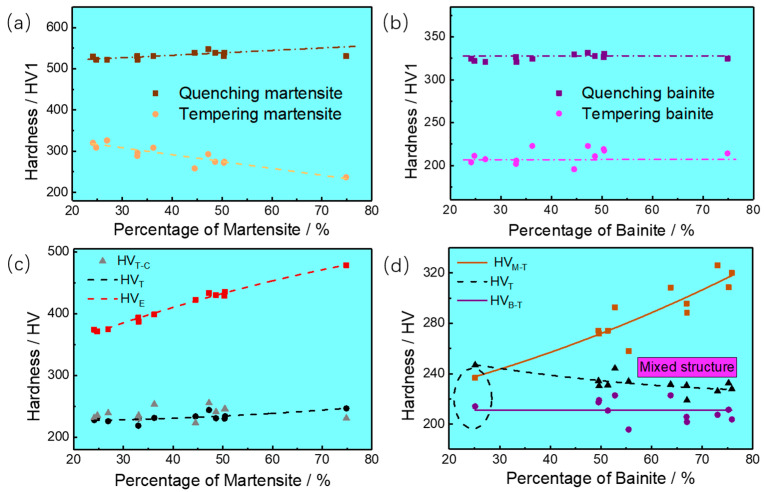
Relationship between the fraction of LM/GB structure and microhardness. (**a**) Martensitic hardness-Percentage of Martensite/%, (**b**) Bainitic hardness-Percentage of Bainite/%, (**c**) Relationshio between macrohardness of steel and Percentage of Martensite, (**d**) Relationship between microhardness and Percentage of Bainite.

**Figure 6 materials-16-04657-f006:**
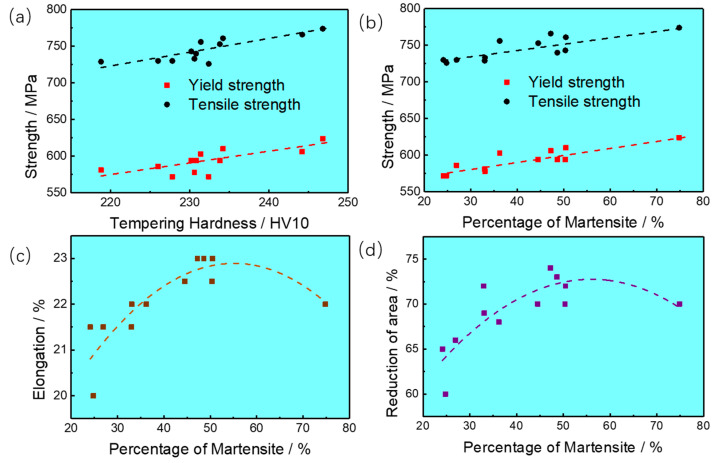
Relationship between microstructure and mechanical properties of experimental steel. (**a**) Relationship between strength and tempering hardness, (**b**) Relationship between strength and percentage of martensite/%, (**c**) Relationship between Elongation and percentage of martensite, (**d**) Relationship between reduction of area and percentage of martensite/%.

**Figure 7 materials-16-04657-f007:**
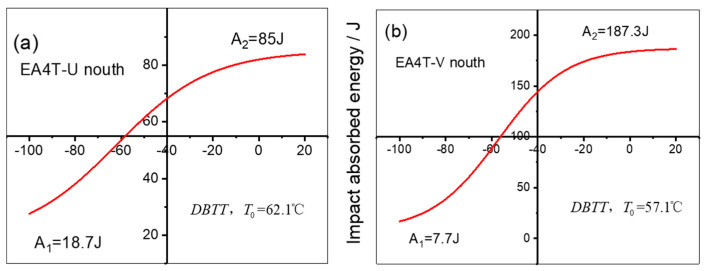
DBTT curves of the samples. (**a**) DBTT curves of CUN, (**b**) DBTT curves of CUV.

**Figure 8 materials-16-04657-f008:**
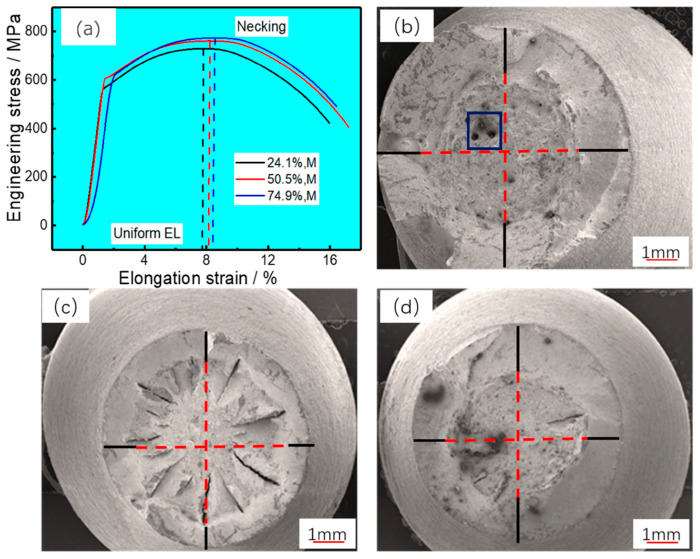
Tensile stress-strain curves and macroscopic fracture morphology of typical specimens. (**a**) stress-strain curves, (**b**) 24.1% martensite, (**c**) 50.5% martensite, and (**d**) 74.9% martensite. The black lines, red lines and pores are the radiating zone, ductile fracture zone and the large dimple, respectively.

**Figure 9 materials-16-04657-f009:**
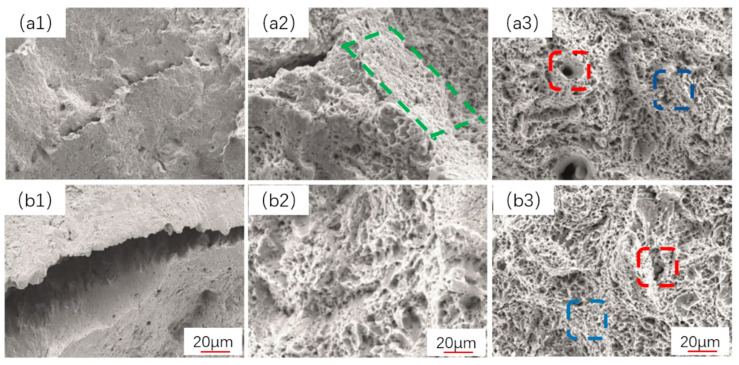
Tensile microscopic fracture morphology observation results: (**a1**–**a3**) 50.5% martensite, (**b1**–**b3**) 74.9% martensite. The green box, red box and blue box are deeper dampers, micro-dimples and the fiber fracture surfaces, respectively.

**Figure 10 materials-16-04657-f010:**
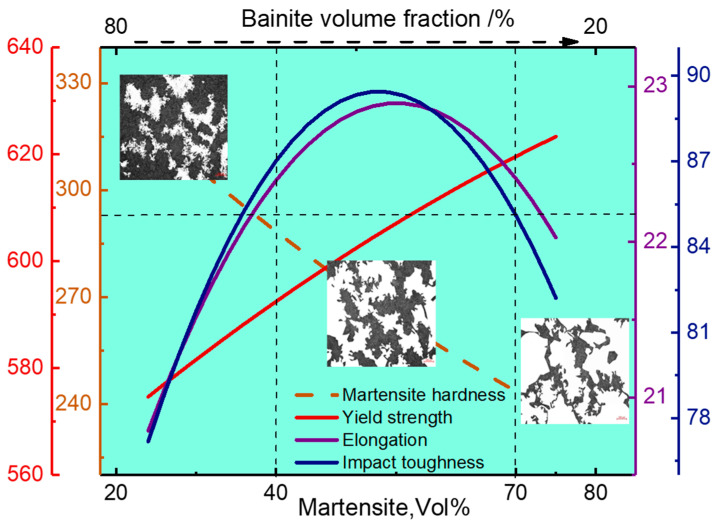
Mechanism of strength, ductility, and impact toughness improvement.

**Table 1 materials-16-04657-t001:** Chemical composition of three specimens.

No.	C	Si	Mn	P	S	Cr	Ni	Cu	V	Mo	W
1#	0.27	0.29	0.72	0.011	0.0014	1.15	0.24	0.035	0.038	0.24	0.004
5#	0.26	0.29	0.70	0.011	0.0019	1.11	0.24	0.034	0.037	0.24	0.004
10#	0.28	0.30	0.74	0.013	0.0025	1.13	0.25	0.041	0.039	0.25	0.004

**Table 2 materials-16-04657-t002:** Martensite/bainite fraction and mechanical properties of EA4T steel.

No.	*M* /%	*B* /%	*HV_T_* /*HV*1	*HV_T-M_* /*HV*1	*HV_T-B_* /*HV*1	*HV_E_* /*HV*1	*HV_E-M_* /*HV*1	*HV_E-B_* /*HV*1	*R_p_*_0.2_ /MPa	*R_m_* /MPa	*A* /%	*Z* /%	*KU*_2_ /J
1#	24.1	75.9	228	320	204	374	530	325	572	730	21.5	65	78
2#	24.8	75.2	232	309	211	371	522	322	572	726	20.0	60	77
3#	27.0	73.0	226	326	207	375	522	321	586	730	21.5	66	82
4#	33.0	67.0	219	288	202	394	531	326	581	729	21.5	72	79
5#	33.1	66.9	231	296	206	387	522	321	578	733	22.0	69	83
6#	36.2	63.8	231	308	223	399	531	324	603	756	22.0	68	88
7#	44.5	55.5	234	258	196	423	539	329	594	753	22.5	70	88
8#	47.2	52.8	244	293	223	433	547	331	606	766	23.0	74	88
9#	48.6	51.4	231	274	211	430	539	328	594	740	23.0	73	89
10#	50.4	49.6	230	272	219	429	531	326	594	743	22.5	70	88
11#	50.5	49.5	234	274	217	436	539	330	610	761	23.0	72	93
12#	74.9	25.1	247	237	214	479	530	324	624	774	22.0	70	82

**Table 3 materials-16-04657-t003:** Prediction accuracy of the mechanical properties model in EA4T steel.

	Equations (9)–(13)		Equations (14)–(18)	
	*ρ*	*δ*/%	*ρ*	*δ*/%
** *R_m_* **	0.90	0.93	0.83	0.93
** *R* ** ** * _eH_ * **	0.92	0.62	0.89	1.02
** *A* **	0.89	2.94	0.88	1.42
**Z**	0.93	1.70	0.83	2.44
** *KU* ** ** _2_ **	0.93	1.63	0.90	1.85

**Table 4 materials-16-04657-t004:** Strength of the martensite and bainite structure.

*M*%	$\sigma_{y}^{M}$	$\sigma_{y}^{B}$	$\sigma_{y}^{M}/\sigma_{y}^{B}$	*HV_T-M_*/*HV_T-B_*
24.1–36.2	766.5	510.3	1.50	1.44–1.57
47.2–50.5	670.0	548.8	1.22	1.26–1.31
50.5–74.9	638.4	581.0	1.10	1.11–1.26

## Data Availability

Not applicable.

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
