# Peer review of "Influence of Martensite/Bainite Dual Phase-Content on the Mechanical Properties of EA4T High-Speed Axle Steel"

_materials, 2023, doi:10.3390/ma16134657_

Round 1
Reviewer 1 Report
Authors of the publication "Influence of martensite/bainite dualphase-content on the mechanical properties of EA4T high-speed axle steel ” presented the relationship between microstructure and mechanical properties in EA4T steel. The authors research on samples with different volume fractions of martensite/bainite, in which the share of individual components of the structure was quantified and characterized using an optical microscope and electron backscatter diffraction technique.
They also examined the micro-hardness, tensile properties, and impact toughness. Through regression analysis of the experimental data, predictive models of mechanical properties were determined by evaluating the martensite/bainite. The researchers also determined volume fraction and the mechanism of the LM/GB structure's impact on strength, plasticity and ductility.
Each of the presented parts of the publication has been correctly described by the authors. The conclusions are consistent and closely related to the research topic. As a reviewer of this work, however, I believe that the reviewed work requires small corrections, which will undoubtedly improve its quality:
1. writing language should be improved. We write impersonally in publications.
2. A digestings reagent should be administered.
3. Linia 89 - "billet→â–¡280mm" - error
4. Figure 2d - unreadable
5. Patterns 1 and 2 - lack ")"
6.Figure 4 - unreadable
7. Line 137 - error - instead of Figure 1 it should be Figure 5
8. Only 10 literature items from the last 5 years.
Author Response
We really appreciate your hard work on the paper and your suggestions for revision.We have made modifications one by one.Thank you very much for your technical guidance and detailed review.

Reviewer 2 Report
1. How does the Martensite /Bainite fraction is obtained.
2. It is suggested to include the Heat Treatment TTT curve.
3.Equations 1,2 no closing brackets, Source of Equations may be cited. Please provide references.
4.Impact tests were done by varying temperatures: Authors may mention the quenching media used.
5. Authors may specify the type of etchant used?
7. kindly confirm whether ferrite is available on the specimen
8.Discuss about the effect of alloying elements, include carbon equivalent Ce
9.Overall Spelling mistakes, Grammar to be improved.
10.Mention the standards for tensile test.
11.Reference no 6, 37 to be rechecked
Spelling mistakes are to be corrected and Grammar is to be improved.
Author Response
We really appreciate your hard work on the paper and your suggestions for revison.We have made modificaitions one by one.Thank you very much for your technical guidance and detailed review.

Reviewer 3 Report
Revision article “influence martensite/bainite dualphase-content on the mechanical properties of EA4T high speed axle steel” The authors aim to analyse the relationship between the mechanical properties and the structure. To achieve this goal, they have used different thermal treatments with the aim of creating structural modifications that affect the relationship between martensite and bainite. The authors have used different techniques to carry out the structural and morphological characterization. In addition, they have carried out different essays to determine the mechanical properties of the different steels obtained. The authors have carried out an exhaustive and laborious study with a notable experimental part. However, from my point of view, there are certain aspects to improve
Format and writing:
It would be convenient for the authors to read the document to detect typographical errors in the text (line 41, line 89, scientific notation of line 126...)
Some sentences are too long and difficult to read or understand (sentences in line 47-48...). I would advise using shorter sentences and using more tables so that the reading is not so cumbersome.
The authors should improve the format of the equations; many go outside the margins of the text.
They should improve the quality of all images. It is difficult to see in the images the explanations given in the text. They do not have enough resolution or they are too small. The figure 1 that comes after the 4, I understand is a mistake. All the graphs in Figure 4 should be sharper and remove all the equations from the interior that detract from the quality of the image. The graphics in figure 6 are perfect but the images cannot appreciate the details needed to follow the information given by the authors
The table format falls outside the page limits.
Questions:
In line 121 it is commented that the chemical composition is obtained but it does not indicate how and what are the results obtained.
In line 204, the authors indicate that they have made an important finding but this is not evident in the conclusions. In reality, the conclusions are poor in relation to the large amount of experimental work carried out.
Author Response
We really appreciate your hard work on the paper and your suggestions for revison.We have made modifications one by one.Thank you very much for your technical guidance and detailed review.

Round 2
Reviewer 2 Report
Accepted